# The Impact of City-Led Neighborhood Action on the Coproduction of Neighborhood Quality and Safety in Buffalo, NY

**DOI:** 10.3390/ijerph22030341

**Published:** 2025-02-26

**Authors:** Katharine Robb, Pablo Uribe, Eleanor Dickens, Ashley Marcoux, Jessica Creighton, Jorrit de Jong

**Affiliations:** Bloomberg Harvard City Leadership Initiative, Harvard Kennedy School, 79 JFK Street, Cambridge, MA 02138, USA; uribe.pablo90@gmail.com (P.U.); egdickens@gmail.com (E.D.); amarcoux9@gmail.com (A.M.); jessica_creighton@hks.harvard.edu (J.C.); jorrit_dejong@hks.harvard.edu (J.d.J.)

**Keywords:** social determinants of health, blight, crime, community engagement, trust, reporting

## Abstract

Creating and sustaining safe, healthy urban environments requires active collaboration between residents and local governments. Public safety and the upkeep of public spaces depend, in a large part, on residents’ reports of crime and service needs. However, in underserved areas, factors such as urban decay, inadequate public services, and concentrated disadvantage have weakened these cooperative dynamics. This breakdown can exacerbate the underreporting of crime and service needs and deepen neighborhood inequalities. In Buffalo, NY, the city-led initiative “Clean Sweeps” works to reduce neighborhood disparities through rapid beautification and community outreach in targeted city blocks. The program aims to improve quality of life by reducing crime and blight while fostering greater community engagement. In an analysis of data from 77,955 matched properties (published elsewhere), we found that residents were more likely to report drug-related crimes (via 911) and blight-related service needs (via 311) compared to untreated properties in the 6 months following the Clean Sweep. In this study, we analyze data from 21 interviews with city staff and four focus groups with residents to explore how interventions in the social and physical environment of neighborhoods, like the Clean Sweep innovation, can influence residents’ willingness to coproduce with local government. We identify improved responsiveness, trust, and self-efficacy as key mechanisms impacting residents’ reporting behavior. The findings show how relatively simple environmental interventions paired with outreach can help create safer, healthier neighborhoods.

## 1. Introduction

Building and maintaining safe and healthy urban environments requires active cooperation between residents and local government [1]. However, urban decay, low-quality public services, and concentrated disadvantages have weakened these cooperative dynamics in underserved areas [2]. Disorder and persistent poverty are associated with feelings of powerlessness among residents, less participation in political activities, and less social cohesion [3,4,5]. Research shows how persistent social problems and deficient public services create an environment that fosters interpersonal violence, legal cynicism, dissatisfaction with the police, and tolerance for deviance [6,7]. Consequently, a local government’s failure to provide adequate public services diminishes its legitimacy and weakens the trust of communities, resulting in alienation and hindering coproduction to produce better outcomes.

In Buffalo, NY, a city-led initiative, called Clean Sweeps, seeks to improve quality of life for residents and repair a fractured relationship between the community and local government through rapid beautification and community outreach in clusters of city blocks. Each week from April through October, city workers from tens of departments, alongside community partners, go door to door in distressed neighborhoods, greeting residents and sharing information on city and community services. Community police officers introduce themselves to residents and listen to their concerns. Meanwhile, other city workers attend to the physical environment: trimming vegetation, removing trash and graffiti, repairing signs, and boarding up vacant properties.

A previous study by Dickens et al. found that in a matched sample of properties from 2009 to 2021 (*n* = 77,955), properties that received a Clean Sweep were 42% more likely, within six months following the Clean Sweep, to report drug-related crimes via 911 and 9% more likely to report blight-related service needs via 311 (e.g., trash removal, vacant property board-up) compared to properties that did not receive a Clean Sweep [8]. No significant difference in the reporting of violent, property, or “other” crime was found, nor was there a difference in the reporting of non-blight-related service needs between properties that received a Clean Sweep and those that did not (see Appendix A).

In this paper, we ask what explains the observed increase in reporting behavior in underinvested neighborhoods after a Clean Sweep. We use qualitative methods to examine the possible causal mechanisms underlying these findings and offer new propositions regarding the association between interventions in the physical and social environment of neighborhoods and residents’ willingness to coproduce with local government to create safer, healthier neighborhoods.

### 1.1. Literature Review

Coproduction is defined as the participation of residents in the design, delivery, or evaluation of public services [9]. By reporting crime and service needs, residents involve themselves in coproduction [10,11,12,13]. Since most illegal behavior is hidden from the authorities’ view, information from the public is essential to successfully investigate and resolve crime and the degradation of public space. Reporting is also necessary for distributing scarce resources effectively, efficiently, and equitably [10,14,15].

In American cities, most crime is not reported. In 2022, only 42% of violent victimizations and 32% of property victimizations were reported to the police [16]. Resident reporting does not represent the underlying magnitude or distribution of crime within a city [17,18]. A variety of situational (e.g., whether the victim was injured), contextual (e.g., the community’s relationship with police), rational (e.g., if the benefits of reporting outweigh costs) and normative (e.g., beliefs about whether the police care) factors influence crime reporting [19]. While crime severity is the strongest predictor of reporting, community-level factors play an important role [20].

Where trust in and the responsiveness of authorities is low, so is reporting behavior [21,22]. Numerous studies show that trust in the police is driven by a belief in procedural justice, which is shaped by policing practices [13,21,23]. Residents of blighted areas often have negative experiences with frontline staff, such as the police and code enforcement officers, who they may only encounter in punitive scenarios [17,21]. Further, physical disorder in neighborhoods gives the impression of low government responsiveness and low social control, signaling that crime may be tolerated and that reporting is ineffectual [5,24,25]. Such areas are often labeled as “problem places” and their residents as “problem people”, amplifying stigmatization and marginalization [26]. This framing contributes to economic disinvestment and the degradation of public services, reinforcing cycles of poverty and exclusion [27]. As a result, residents may become more disengaged and less likely to report problems, feeling alienated from the institutions meant to serve them [28].

Drug crimes are among the most underreported types of crimes [29,30]. Reporting a drug crime can be costly—in terms of physical or psychological safety—and the cost–benefit analysis often means people do not carry it out [31]. If residents do not feel safe from retaliation, they will not voluntarily report drug crimes. Further, unlike violent or property crimes, drug crimes do not have a clear, direct victim. Their negative effects are dispersed across communities, and it is not clear who should report them.

Like crime reporting, calls to 311 similarly under-represent the needs of residents [32]. The non-emergency number, 311, connects residents to local government services in many U.S. and Canadian cities. It is primarily used to report problems (e.g., noise complaints, graffiti, illegal dumping) and request services (e.g., tree trimming, pothole repair). However, many residents are unwilling, unable, or unaware that they can report service needs, presenting socio-spatial disparities in 311 reporting across cities [33]. Similar to 911, 311 use is similarly influenced by residents’ trust in the capacity and will of the government to respond in a just way [34]. Like drug crime, blight-related service needs often impact more than just the person who reports the problem [35]. Therefore, reporting responsibility is not always clear and represents a time cost to the reporter. Furthermore, communities with a high concentration of physical disorder are less likely to take collective action to address blight, making them less likely to report their needs [25,36,37,38].

Drug crimes and blight generate an array of collective problems for communities, as well as a range of negative effects on health. Areas where drug activity is concentrated experience higher levels of property and violent crime, as well as greater physical disorder [37,39,40]. The crime and disorder that drug activity attracts generates other negative social determinants of health such as lower educational attainment [41], lower private investment [42], fewer jobs [43], social isolation [44], and decreased physical activity [45]. Drug crime and blight are associated with a wide range mental and physical health outcomes—from all-cause premature mortality to depression and diabetes [36,46,47].

However, reporting behaviors are responsive to shifts in both the physical and social environment of neighborhoods [12,20,48]. A growing body of evidence supports the effectiveness of collaborative, multi-agency environmental and outreach approaches that do not necessarily involve the police [49]. These strategies leverage diverse expertise within and outside of local government to address to a complex web of risk factors that allow crime and blight to remain persistently high in distressed neighborhoods. The Buffalo Clean Sweep Initiative brings together diverse actors to engage in blight reduction and community outreach in high-need city blocks. These activities may influence residents’ reporting of crime and service needs through a variety of mechanisms.

### 1.2. Objective

Building on quantitative findings that link the Clean Sweep Initiative to the increased reporting of drug-related crime and blight-related service needs [8], the animating research question of this work is as follows: what explains the observed increase in reporting behavior following the Clean Sweep? Using qualitative data from interviews and focus groups with city staff and residents in Buffalo, NY, we explore the mechanisms through which city-led, place-based interventions, like the Clean Sweep, influence the coproduction of public safety and public space.

## 2. Materials and Methods

### 2.1. Case Context

Buffalo is the second largest city in New York state. It is a “Rust Belt” city, synonymous with former manufacturing epicenters confronting the socioeconomic fallout of deindustrialization, population loss, and shrinking municipal budgets [50]. Buffalo shares its current social and economic challenges with other U.S. and international cities where population decline, economic downturn, rising unemployment, structural racism, and other social problems have led to neighborhood neglect and abandonment [51].

Today, 28% of Buffalonians live at or below the poverty line and racial disparities are staggering. Nearly 70% of Black residents live in poverty compared to 20% of White residents [52], a legacy of red-lining and other forms of structural racism [50]. Most of Buffalo’s residents are renters (60%), reflective of systemic barriers to wealth accumulation. Physical signs of vacant and abandoned properties abound—17% of properties in the city are vacant, and on Buffalo’s East side, the vacancy rate is 27% [52].

While the City of Buffalo has made strides in improving police–community relationships through several initiatives—including community policing and increasing the diversity of the police force—the city faces deep-seated challenges. As was the case in cities across the U.S., from the “War on Drugs” beginning in the 1970s through the “War on Terror” beginning in 2001, the predominate style of policing encouraged an aggressive response to quality-of-life crimes and resulted in estrangement from communities and high arrest and incarceration rates in areas of concentrated poverty [53]. In a 2016 report from Buffalo’s Community Policing Survey, only 44% of Black respondents believed they could trust the police, compared to 72% of White respondents [53].

### 2.2. The Clean Sweep Initiative

For over 20 years, the City of Buffalo, NY, has brought city staff and community partners together for targeted and rapid beautification and community outreach in neighborhoods impacted by disinvestment and structural inequities. The goal is to enhance quality of life by reducing crime and blight and improving community engagement. Every Wednesday from April through October, city staff from more than ten departments and tens of community organizations work in clusters of around 200 properties, encompassing an area of two to three city blocks. City staff and community organization representatives go door to door, greeting residents and handing out bags containing information on city and community services, including healthcare, home improvement, the city’s non-emergency helpline (311), and more. If no one is home, they leave the bag on the porch. If a resident opens the door, they engage the resident in conversation to learn about their needs and listen to their concerns. Meanwhile, public works and forestry teams trim trees, grass, and bushes; remove graffiti and trash; repair signs and sewer grates; and board up vacant properties. Housing inspectors inspect the exteriors of homes. Community police officers introduce themselves to residents, listen to concerns, and receive reports about local criminal activity. They also make residents aware of public safety programs and ways to report tips anonymously. Other city staff are on hand to answer questions from residents. Through this process, which mobilizes resources and creativity within and outside the city hall, the city is able to overcome common restrictions of siloed work and inadequate resources, as described in Robb et al. [54]. In contrast to more siloed approaches to blight reduction and outreach, the Clean Sweep’s approach is problem-oriented and collaborative; each Clean Sweep offers opportunities for government officials and community partners to build capabilities to address the problems they encounter [54,55].

### 2.3. Data Collection

This study draws on data collected in October 2021, comprising 21 semi-structured key informant interviews (KIIs) with city staff and four focus group discussions (FGDs) with residents whose neighborhoods had received a Clean Sweep. Participants were asked to reflect on whether and why the goals of the Clean Sweep were met, as well as how the initiative was perceived across diverse community groups. KIIs were led by the first, third, and fourth authors. FGDs were led by Buffalo community leaders trained as facilitators by the research team. City staff (*n* = 13) were recruited through purposive and snowball sampling and interviews were held at the city hall or on Zoom, lasting 60–90 min each. Eight KIIs with city staff were conducted during a Clean Sweep, lasting 5–15 min each. Four FGDs were held with residents whose blocks had received a Clean Sweep in the last several years (16 total participants). Residents were recruited by flyers placed in neighborhoods and email lists maintained by block clubs and Buffalo’s Department of Citizen Services. The FGDs occurred in the evening at community centers at four different locations in the city and lasted 90 min. The racial composition of focus group participants was 75% Black, 12.5% White, and 12.5% Latino/a. Women made up a larger proportion (68.8%) than men (31.2%). The age distribution included 31% of participants age 18–24; 38% age 25–54; and 31% age 55 or older.

The data, particularly the FGD data, are subject to several biases, including social desirability and selection bias. To mitigate bias, respondents were asked to share viewpoints of which they were aware but did not hear reflected in the discussion and were encouraged to think about how other residents might respond to the discussion questions. This was particularly important given that the recruitment strategy may have disproportionately attracted people who were already active in their communities or with the local government.

All KIIs and FGDs were recorded and transcribed. Thematic analysis was used to code transcripts in Dedoose (version 8.0.35). Each transcript was coded independently by two researchers and discrepancies were discussed and resolved through deliberation. The codes included deductive codes from the literature and inductive codes from the data. Sub-themes were identified after the initial coding, and a codebook was developed that contained 14 parent codes and 26 child codes. Example codes include “change in reported crime”, “change in actual crime”, “types of crime”, “blight”, “311 use”, “trust”, “self-efficacy”, “collective efficacy”, “perceptions of government authorities”, “responsiveness”, and “perceptions of Clean Sweeps”.

## 3. Results

The thematic analysis of focus groups with residents and interviews with city staff revealed three primary mechanisms shaping residents’ participation in the coproduction of neighborhood safety and quality. These include (1) visible responsiveness from local government officials to residents’ concerns, (2) the building of trust between community members and the local government, and (3) the activation of residents’ self-efficacy in countering social and physical disorder.

### 3.1. Responsiveness

A recurring theme in discussions with residents was the sense of being neglected or abandoned by the city. Many expressed frustration with the infrequent presence of police and government officials, noting that engagement typically occurred only during election seasons or in response to punitive situations. Residents also highlighted the multiple barriers they faced when attempting to navigate municipal services, describing the process as leaving them feeling disenfranchised. One resident summarized the relationship between city authorities and the community as “There’s no back-and-forth communication. We don’t have access to resources, or we don’t know how to access them”. Another resident explained, “It’s like we don’t exist. We’re beat down. There’s anger towards this system. I understand why people toss garbage out the window or dump in an empty lot—because there’s no connection, there’s no feeling that somebody cares”.

However, seeing the city take an interest in their neighborhood and make visible improvements during the Clean Sweep modified residents’ perceptions of city authorities. “It brings the message that we’re important”, stated one resident, “that our place is important to the city”. Residents described returning home to a street that looked markedly different, with extensive or complete reductions in accumulated trash, overgrown lots, and broken signage. When describing conditions after the Clean Sweep, one resident said, “When everybody got home to see it, they were excited. And maybe it made them feel better about themselves. They kept it up from there”. Another resident said “[The Clean Sweep] made me feel like the city wanted us to know that they had our best interest in mind and that the city would assist us”.

Still, not all residents were convinced. Many expressed that more sustained efforts by the city would be required to change attitudes. One resident noted that the Clean Sweep was “just a good gesture”, while another shared that “having the impetus to walk through the neighborhoods and try to develop a rapport is one thing, but when the city isn’t following through with other things, people become disillusioned”.

City staff acknowledged the fractured relationship between the local government and residents, recognizing the negative perceptions and experiences residents had encountered in their interactions with local authorities. However, they also highlighted numerous instances of being able to respond to residents’ needs in new and impactful ways. A city employee explained the following:


*Especially in some of the lower socioeconomic areas, people feel there’s a distance between them and their government. When they can see direct impact of the services initiated by us after a conversation we’ve had with them—you can see that relief on their faces. It’s almost like a refresh button. There is something very visceral about seeing your needs addressed right there on the spot.*


Similarly, a police officer described the following:


*The [Clean Sweep] goes to areas that have traditionally had issues with quality of life. A field that was overgrown that had crimes happening in it—[residents] visibly see it being cut down and cleaned up. And they respond to that, especially when it’s something that’s really frustrating to them and they’re dealing with the crime associated with it.*


In addition to the impact of changes to the environment, another police officer described the impact of changes to social relationships:


*[Clean Sweeps] gives us police officers a chance to really sit there and talk to [residents], to listen and try to sort out their issues, one by one. And sometimes deal with it directly. I’ve called police off calls and pulled them over to say, ‘Hey, this place—I need you to take care of this right now.’ And that means a lot to people. It makes people feel like you care and that they’re a priority.*


While some concerns from residents persisted, both residents and city staff overwhelmingly reported that visible improvements to the physical environment and prompt responses to community concerns significantly enhanced residents’ perceptions of government responsiveness and attention to their concerns.

### 3.2. Trust

Prior to the Clean Sweep, residents’ feelings and experiences of abandonment and unfair treatment weakened trust in and the legitimacy of the police and city government. As a resident remarked, “The city needs to reach out to the people not just with enforcement, there has to be a reaching out in trust”. Both city staff and residents reported that the city authorities’ respectful and just interactions during Clean Sweeps helped establish or re-establish legitimacy with residents—who in turn provided authorities with critical information, particularly related to drug crime. The structure of the Clean Sweep, including door-to-door outreach, created opportunities for meaningful interactions that, according to city staff, are often not possible outside of Clean Sweep settings. During these interactions, city staff and police officers devote considerable time and effort to listen and respond to residents’ concerns and needs. One police officer noted,


*I’ve found the more positive interactions and more available we are, the more there’s a sense of safety and people’s bravery [to report crime] increases. One resident [during the Clean Sweep] told me about a drug operation on her street. We knew nothing of it; there had been no reports. From what she was saying, people were very afraid on the street, and no one was calling about it. I transferred it to our investigative unit and they’re working on it now.*


Another police officer described how working alongside other city and community staff made it possible for residents to feel more comfortable talking with him. Without the ability to have these types of conversations, he would not be able to share and receive critical information.


*[During the Clean Sweep] when I come to the door, being that I’m surrounded by a lot of civilians, it looks normal. I’m not over there as an investigator for a 911 call. It gives me the privacy to say, ‘Hey, do you know about our anonymous tip line?’ When the community sees that what I’m trying to do is crime prevention, they say, ‘Yeah, you know what? Now that you mention it, this house, that house [pointing out houses with drug activity].’*


Still, not all residents saw the Clean Sweep as increasing trust. One summarized a common critique, stating “A one-day thing, it doesn’t change mindset”. Yet as a city staff person described,


*I see officers changing that narrative (of lack of trust in city officials) every time we go on a Clean Sweep […] Coming out alongside the teams doing the hands-on work to beautify the neighborhood, that builds trust. It builds trust enough to open the door—to literally open the door.*


Overall, interviewees indicated that the outreach component of the Clean Sweep facilitated interactions that helped restore or create trust between residents and government, thereby increasing residents’ willingness to engage with city authorities under the expectation of fair treatment and outcomes.

### 3.3. Self-Efficacy

Residents and city staff described the impact that the Clean Sweep had on residents’ willingness to engage individually, as a community, and with government in maintaining a cleaner and safer neighborhood. One resident described the impact of the Clean Sweep as, “We know what we need to do but sometimes we need a push—the Clean Sweep is a little push”. When asked about how people’s behaviors in the neighborhood changed following the Clean Sweep, one resident stated, “People just started taking care of their stuff a little bit more. We stayed on top of it being clean”. Some residents described how observing government responsiveness through highly visible blight reduction showed them what was possible. This was typified by one resident stating “I think [the impact of blight remediation] lasts because people now see that it can be improved, that was achievable, and it can be maintained”. This sentiment was further exemplified by another resident who stated,


*Now people feel like they can sit on their porch and that reduces random people from just coming down and parking and exchanging drugs. We chase it out by setting a standard. It’s a standard set when Clean Sweep comes to an area.*


City staff also depicted increases in residents’ self-efficacy. A city employee said,


*There are people I’ve met on Clean Sweeps who didn’t know about 311 and are now taking full advantage. They’re calling, they know who to talk to, what department to go to. Those are the changes that I think matter most. Those are the ones that seem to last.*


In addition to describing increases in 311 usage, police officers also noted a rise in call volumes following a Clean Sweep. One officer suggested, “I think that’s because people feel like they actually have more access, they feel more comfortable after making those contacts [with police officers]”.

In addition to the increased volume of calls to city officials, both city staff and residents reported other notable increases in resident-led initiatives aimed at maintaining public safety and public spaces. For instance, when asked about the influence of the Clean Sweep on residents’ behavior, one resident noted, “As a matter of fact, yes. We started a block club”. Overall, the interviews and focus groups revealed multiple instances in which improvements to the physical environment, alongside positive interactions between residents and city officials, enhanced perceptions of government responsiveness and built trust. In turn, this contributed to improvements in residents’ sense of self-efficacy in improving neighborhood safety and quality of life.

## 4. Discussion

The focus group and interview data reveal how a city-led beautification and community outreach initiative (the Clean Sweep) can improve residents’ perceptions of government responsiveness, build trust between residents and government, and increase residents’ self-efficacy. These changes, brought about through intervention in the physical and social environment, resulted in residents being more willing to engage in coproduction with local government. This was evidenced both by the findings of Dickens et al. of increased reporting via 911 and 311 [8] and in the present qualitative study. The matched sample quantitative approach used by Dickens et al. identified an increase in the reporting of drug-related crime and blight-related service and no changes in other types of crime or service needs following the Clean Sweep; however, the study could only speculate about the underlying mechanisms. Complex real-world phenomena, such as changes in reporting behavior, often involve causal pathways that cannot be captured by a single methodological approach [56]. Through qualitative analysis, this study provides a rich description from multiple perspectives on how Clean Sweep activities may influence reporting behaviors, offering contextual insights into the mechanisms driving these changes.

As described in the introduction, safe and healthy urban environments depend on a partnership between residents and local governments [1]. By promptly acknowledging and acting upon residents’ inputs and concerns, governments can rebuild trust and shared ownership [9,57,58]. Yet, a lack of government responsiveness, especially in areas of historic neglect and marginalization, exacerbates inequalities and further erodes the relationship between the government and its citizens [2,18,59]. This sentiment was reflected in the FGDs and KIIs as “distance” and “no trust” between residents and government. Residents describing feeling “beat down” and as if they “don’t exist” signaled a fractured relationship between residents and government authorities.

In this setting, the Clean Sweep provided a highly visible demonstration of what the combined efforts of multiple government departments can achieve. The intervention showed what the government can and is willing to do for residents, helping to break a pattern that has been observed for generations in underserved areas. When citizens see the government as a responsive actor, they are more willing to trust and collaborate [57,58]. Following a Clean Sweep, residents described feeling as though “the city cares” and that their “place is important to the city”—sentiments that play a pivotal role in enhancing resident self-efficacy [60,61].

Residents of highly blighted areas often perceive that they are not able to effect change in their surroundings, much less have an influence on their neighbors or government officials [62,63]. Previous research has shown that citizen self-efficacy is influenced by the extent to which the government is responsive and respectful [58,64] and citizens’ level of trust in the government [60]. In low-trust environments, citizens may view coproduction with the government, for example, reporting a crime, as putting themselves in too vulnerable a position and without the expectation of a positive outcome [12,17,30,65].

A seminal study by Tyler and Fagan [13] found that willingness to cooperate with authorities was higher among respondents who perceived those authorities as more legitimate—and that people see authorities as more legitimate if they make decisions fairly and treat them fairly. Residents and city staff described increased willingness among residents to engage with local government, having seen that government authorities can respond in fair and positive ways. City staff described the Clean Sweep as a “refresh button”, opening the door for conversations between residents and city staff and increasing “bravery” among residents to report incidents. As described by residents, the Clean Sweep activities helped to “set a standard” for what is possible—that government can be responsive and trustworthy and that residents’ actions matter. As found in our study and the previous literature, residents are also more likely to coproduce by taking direct actions to prevent unwanted social behavior or the physical deterioration of public spaces on their own (e.g., forming neighborhood block clubs) when government responsiveness and trust in the government are high [12,13].

Finally, at first glance, the increased reporting of drug crime and blight might suggest that the intervention contributed to neighborhood decline. However, neither the qualitative data nor prior research support this interpretation. As outlined in the introduction, actual crime and service needs are different measures than reported crime and service needs. Reporting behaviors are influenced by various factors, including the social and environmental context, the nature of the incident, and perceived costs and benefits of reporting. The Clean Sweep initiative’s main activities are beautification and community engagement. Therefore, it is reasonable that its impact was observed in an increase in reports of physical and social disorder and not in violent, property, or “other” crime or in other types of service requests not demonstrated during the Clean Sweep [57,58].

The study findings reinforce the importance of interventions by local governments and their partners that actively work to dismantle the alienation and mistrust between residents and public officials. Future research should continue to explore how improving responsiveness, trust, and resident self-efficacy can help enrich areas of concentrated disadvantage, build cooperative dynamics, and improve social outcomes. Future research also should examine how these mechanisms impact coproductive behaviors in settings with weaker or stronger relationships between government and residents. Further, a more nuanced understanding of how these mechanisms function individually, synergistically, and over time could aid in the design of programs that facilitate greater resident engagement and improved neighborhood conditions. Finally, future research should also examine the impact of improving cooperative dynamics between city officials and residents on critical outcomes such as stress, violence, physical activity, justice system involvement, education, and other social determinants of health.

## 5. Conclusions

In this paper, we explored factors that account for the observed increase in the reporting of drug-related crime and blight-related service needs following a city-led beautification and community outreach intervention (the Clean Sweep). Through our analysis, we put forth three mechanisms: (1) an increase in residents’ perceptions of government responsiveness, (2) the building of trust between residents and government, and (3) an increase in residents’ self-efficacy.

By leveraging expertise and resources from within and outside city hall, the Clean Sweep demonstrates how targeted interventions can increase residents’ willingness to coproduce with local government. Public authorities can bridge gaps between local government and underserved communities by enhancing responsiveness, trust, and self-efficacy. Key recommendations include proactively addressing physical disorder to ensure visible improvements for residents; implementing collaborative interventions that pair enforcement with service provision to position local authorities as partners; and investing in community engagement to facilitate meaningful dialog between residents and city officials.

As urban challenges evolve, understanding the mechanisms driving coproductive behaviors is crucial for designing adaptive and inclusive interventions that enhance community well-being and government effectiveness. By repairing and reinforcing relationships between residents and city governments, such interventions have the potential to generate healthier and safer environments for urban communities, realizing the mutual benefits of resident–government collaboration.

## Data Availability

Deidentified excerpts from transcripts are available upon written request to the corresponding author.

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
