# Peer review of "The Impact of City-Led Neighborhood Action on the Coproduction of Neighborhood Quality and Safety in Buffalo, NY"

_ijerph, 2025, doi:10.3390/ijerph22030341_

Round 1
Reviewer 1 Report
Comments and Suggestions for Authors
This is a really excellent article on the impact of city-led nieghbourhood action around neighborhood quality and safety. The paper presents more nuanced and in-depth qualitative findings building on previously published quantitative findings. These seems very complimentary, and offers a different lens for the reader about the Clean Sweep initiative.
The overall structure, objectives and narrative are very clear and it is easy for the reader to connect to the paper and the main findings.
What was perhaps missing from the literature and the later discussion was consideration of Urban Stigmatisation (following Loic Wacquant's work) that allows critical engagement with the complexity of the process and marginalisation that these communities will experience. This also allows insight to the unfair process of labelling 'problem places, and therefore problem people' (see Gerry Mooney). This literature may be of real interest I think to this topic and discussion?
Overall, however, I thought this was a polished, interesting paper that gives original insights.
Author Response
What was perhaps missing from the literature and the later discussion was consideration of Urban Stigmatisation (following Loic Wacquant's work) that allows critical engagement with the complexity of the process and marginalisation that these communities will experience. This also allows insight to the unfair process of labelling 'problem places, and therefore problem people' (see Gerry Mooney). This literature may be of real interest I think to this topic and discussion?
Thank you for the thoughtful and encouraging feedback. Your recommendation to incorporate literature on urban stigmatization by Loic Wacquant and Gerry Mooney is well taken, and we have integrated these perspectives into both the literature review and the discussion.
Lit review (line 83): “Such areas are often labeled as "problem places," and their residents as "problem people," amplifying stigmatization and marginalization [26]. This framing contributes to economic disinvestment and a degradation of public services, reinforcing cycles of poverty and exclusion [27]. As a result, residents may become more disengaged and less likely to report problems, feeling alienated from the institutions meant to serve them [28].”
Discussion (line 408): “The study findings reinforce the importance of interventions by local governments and their partners that actively work to dismantle the alienation and mistrust between residents and public officials. Future research should continue to explore how improving responsiveness, trust, and resident self-efficacy can help enrich areas of concentrated disadvantage, build cooperative dynamics, and improve social outcomes.”
Reviewer 2 Report
Comments and Suggestions for Authors
The article is well written, but few points should be taken into consideration:
1. there are no hypotheses and no research questions; research questions should be specifically addressed and they could serve as basis to develop hypotheses;
2. data has been collected in October 2021; there are no mentions regarding the changes that could intervene meanwhile;
3. the conclusions should be more elaborated, making reference to hypotheses, research questions and more specifically to the key mechanisms;
4. the conclusion section should include implications derived from this experience and how can this be generalized to the global context; also, some recommendations for public authorities would be valuable.
Author Response
Comment 1:
- there are no hypotheses and no research questions; research questions should be specifically addressed and they could serve as basis to develop hypotheses;
Thank you for this comment and the opportunity to clarify our research question. We have modified the objective section of our paper to more clearly articulate the research question.
(Line 106) "Building on quantitative findings that link the Clean Sweep Initiative to increased reporting of drug-related crime and blight-related service needs [8], the animating research question of this work is: what explains the observed increase in reporting behavior following the Clean Sweep? Using qualitative data from interviews and focus groups with city staff and residents in Buffalo, NY, we explore the mechanisms through which city-led, place-based interventions, like the Clean Sweep, influence the coproduction of public safety and public space.
As the paper is an exploratory, qualitative study, we did not set out with specific hypotheses in mind (Creswell & Poth, 2021). Instead, the aim was to explore mechanisms that may influence reporting behavior. Through the analysis, we identified three key mechanisms: responsiveness, trust, and self-efficacy. In subsequent work, these proposed mechanisms could be tested as hypotheses. At the end of the discussion section, we suggested avenues for future research to examine these mechanisms in various contexts, exploring their individual and synergistic functions over time.
Creswell, J. W., & Poth, C. N. (2018). Qualitative Inquiry & Research Design: Choosing Among Five Approaches. Sage Publications.
Comment 2:
- data has been collected in October 2021; there are no mentions regarding the changes that could intervene meanwhile;
The quantitative data span from 2009 to 2021, with qualitative data collected in October 2021.
How I understand the nature of the question is whether events that happened after the study period might influence the findings. Put another way, if we had done our data collection in 2024 might the mechanisms we identified be different?
While we do not have reason to suspect significant changes in the responses of residents and city staff, this is always possible. Clean Sweep work continued as usual through the end of the 2024 season and will continue in 2025. Focus group participants in 2021 were Buffalo residents whose blocks had received a Clean Sweep between 2017 and 2021. Changes that might have occurred after 2021 are not captured in our study.
If I’ve not understood or responded to the nature of the question, I am happy to return to this point.
Commnet 3:
- the conclusions should be more elaborated, making reference to hypotheses, research questions and more specifically to the key mechanisms;
Thank you for your valuable feedback. We have expanded the conclusion section to more directly address the research question and highlight key mechanisms. Please see the updates in the comments on point 4.
Comment 4.
- the conclusion section should include implications derived from this experience and how can this be generalized to the global context; also, some recommendations for public authorities would be valuable.
Thank you for this helpful suggestion. We have updated the conclusion to include implications and recommendations.
(Line 424) “In this paper, we explored factors that account for the observed increase in reporting of drug-related crime and blight-related service needs following a city-led beautification and community outreach intervention (The Clean Sweep). Through our analysis, we put forth three mechanisms: 1) an increase in residents’ perceptions of government responsiveness, 2) the building of trust between residents and government, and 3) an increase residents’ self-efficacy.
By leveraging expertise and resources from within and outside city hall, the Clean Sweep demonstrates how targeted interventions can increase residents’ willingness to coproduce with local government. Public authorities can bridge gaps between local government and underserved communities by enhancing responsiveness, trust, and self-efficacy. Key recommendations include proactively addressing physical disorder to ensure visible improvements for residents; implementing collaborative interventions that pair enforcement with service provision to position local authorities as partners; and investing in community engagement to facilitate meaningful dialogue between residents and city officials.
As urban challenges evolve, understanding the mechanisms driving co-productive behaviors is crucial for designing adaptive and inclusive interventions that enhance community well-being and government effectiveness. By repairing and reinforcing relationships between residents and city governments, such interventions have the potential to generate healthier and safer environments for urban communities, realizing the mutual benefits of resident-government collaboration.”
Round 2
Reviewer 2 Report
Comments and Suggestions for Authors
Happy to see that the authors made the changes accordingly.